# Four Types of TiO_2_ Reduced the Growth of Selected Lactic Acid Bacteria Strains

**DOI:** 10.3390/foods10050939

**Published:** 2021-04-25

**Authors:** Ewa Baranowska-Wójcik, Klaudia Gustaw, Dominik Szwajgier, Patryk Oleszczuk, Bożena Pawlikowska-Pawlęga, Jarosław Pawelec, Justyna Kapral-Piotrowska

**Affiliations:** 1Department of Biotechnology, Microbiology and Human Nutrition, University of Life Sciences, Skromna Street 8, 20-704 Lublin, Poland; klaudia.gustaw@up.lublin.pl; 2Department of Radiochemistry and Environmental Chemistry, Faculty of Chemistry, Maria Curie-Skłodowska University, 20-031 Lublin, Poland; patryk.oleszczuk@poczta.umcs.lublin.pl; 3Department of Functional Anatomy and Cytobiology, Faculty of Biology and Biotechnology, Institute of Biological Sciences, Maria Curie-Skłodowska University, Akademicka 19, 20-033 Lublin, Poland; bozena.pawlikowska-pawlega@poczta.umcs.lublin.pl (B.P.-P.); justyna.kapral-piotrowska@poczta.umcs.lublin.pl (J.K.-P.); 4Institute Microscopic Laboratory, Faculty of Biology and Biotechnology, Institute of Biological Sciences, Maria Curie-Skłodowska University, Akademicka 19, 20-033 Lublin, Poland; jaroslaw.pawelec@poczta.umcs.lublin.pl

**Keywords:** TiO_2_ NPs, nanoparticles, E171, bacterial, microbiome

## Abstract

Food-grade titanium dioxide (TiO_2_) containing a nanoparticle fraction (TiO_2_ NPs -nanoparticles) is widely used as a food additive (E171 in the EU). In recent years, it has increasingly been raising controversies as to the presence or absence of its harmful effects on the gastrointestinal microbiota. The complexity and variability of microbiota species present in the human gastrointestinal tract impede the assessment of the impact of food additives on this ecosystem. As unicellular organisms, bacteria are a very convenient research model for investigation of the toxicity of nanoparticles. We examined the effect of TiO_2_ (three types of food-grade E171 and one TiO_2_ NPs, 21 nm) on the growth of 17 strains of lactic acid bacteria colonizing the human digestive tract. Each bacterial strain was treated with TiO_2_ at four concentrations (60, 150, 300, and 600 mg/L TiO_2_). The differences in the growth of the individual strains were caused by the type and concentration of TiO_2_. It was shown that the growth of a majority of the analyzed strains was decreased by the application of E171 and TiO_2_ NPs already at the concentration of 150 and 300 mg/L. At the highest dose (600 mg/L) of the nanoparticles, the reactions of the bacteria to the different TiO_2_ types used in the experiment varied.

## 1. Introduction

Food additives are widely used in the food industry to improve the flavor, smell, color, and shelf life of food [1]. Food-grade TiO_2_ (E171) is a white pigment and brightening agent used in substantial amounts in confectionery, white sauce, and frosting [2,3]. It should be noted that E171 contains different sized TiO_2_ particles, including nanoparticles <100 nm. Dudefoi et al. [4] detected a level of 17–36% of TiO_2_ NPs (NPs—nanoparticles) in seven purchased food samples. Similarly, in their research, other authors showed the content of the <100 nm TiO_2_ nanoparticle fraction in the range from 10 to 49% [3,5,6], whereas a recent a study conducted by Geiss et al. [7] demonstrated that the level of nanoparticles exceeded 50%. It has been reported that the pigment is contained in over 900 food products worldwide. The daily consumption of the compound depends on the age, body weight, and place of residence. For instance, 0.2–0.7 mg and 1 mg TiO_2_/kg body weight (b.w.) are consumed per day in the USA and Great Britain, respectively. However, due to their lower body weight and the higher consumption of sweets, children under 10 years of age may ingest from 1–2 mg to 2–3 mg TiO_2_/kg per day in the USA and Great Britain, respectively [2,8]. In turn, 0.5–0.7 mg TiO_2_/kg b.w./day [9] and 0.2–0.4 mg TiO_2_/kg b.w./day are consumed in the Netherlands and Europe, respectively [10].

In recent years, the use of titanium dioxide as a food additive (E171) has raised considerable controversy [11,12,13]. For instance, France was the first country to prohibit the use of this food additive for fear of its potential harmfulness [11]. Its presence is increasingly often associated with disorders of the intestinal barrier, including intestinal dysbiosis [4,14,15]. Exposure to the compound may induce chronic changes in the composition and/or metabolic activity of commensal bacteria (intestinal dysbiosis) that exert an effect on the immune system [16]. The bacteria may come into contact with TiO_2_ NPs through both food consumption and intestinal passage, which may affect the host’s microbiota and health [17,18]. Changes in the intestinal microbiota can be induced by stress or inadequate diet and may be associated with such diseases as obesity, inflammatory bowel disease, and diabetes [19]. 

The microbiota in the gastrointestinal tract plays an especially important role as a basis of the health of the host. The commensal microbial community not only contributes to the digestion of dietary fiber but also interacts strongly with epithelial cells to maintain an effective gut barrier separating the organism from the external environment of the host [16,20]. The small size of NPs allows them to cross the cell barrier in the gastrointestinal tract or the mucus layer [21,22].

Determination of the toxicity of TiO_2_ NPs may be influenced by interfacial electrostatic interactions and the physicochemical parameters of the medium (pH, size, ionic strength, temperature, electrolyte composition, and light exposure) [23,24]. There are only few studies assessing the interactions between NPs and gut microbiota; they are mainly focused on direct interactions with intestinal epithelial cells [25,26].

The aim of the present study was to assess the response of selected lactic acid bacteria strains to E171/TiO_2_ NPs, depending on their concentration, size, and applicability.

## 2. Materials and Methods 

### 2.1. Nanoparticles

The investigations were carried out with the use of four types of TiO_2_. Food-grade TiO_2_ (E171) was purchased from three suppliers from Poland: Warchem Sp z o.o., Marki; Biomus, Lublin; and Food Colors, Piotrków Trybunalski (No. 1, 2, and 3, respectively). For comparison, TiO_2_ NPs were purchased from Sigma-Aldrich (CAS Number: 718467-100G. Titanium (IV) oxide, nanopowder, 21 nm) (No. 4) (Figure 1).

### 2.2. Sample Preparation 

Aqueous solutions of each type of TiO_2_ were prepared in glass bottles with deionized water at the concentrations of 60, 150, 300, and 600 mg/L (a, b, c, and d, respectively). Subsequently, each sample was sonicated for 30 min. in an ultrasonic bath (25 °C, 250 W, 50 Hz). Fresh solutions were prepared before each experiment.

### 2.3. NPs Characterization 

#### 2.3.1. Zeta-Potential Measurements

The zeta potential (ζ) of the titanium oxide samples (1–4) was measured using the light scattering method with a Zetasizer 3000 instrument (Malvern, UK). The Zeta Potential measurements were conducted at a pH range from 2.0 to 10.0 with a fixed scattering angle of 90°. Smoluchowski’s equation was used to convert the electric mobility into ζ. Before analysis, water suspensions of the examined samples were prepared by sonification of 5 mg/mL of distilled water for 30 min. 

#### 2.3.2. Microscopy Analysis

The size (morphology) of the nanoparticles was examined using a TEM transmission electron microscope (FEI Tecnai G2 T20 X-Twin Ltd., Japan). The following procedure was employed for preparation of samples for the determinations: a nanoparticle suspension (sonicated to re-suspend the sediment) was pipetted onto formvar/carbon film-coated TEM grids. After 5 min, a drop of the preparation was drained with a piece of filter paper, and the mesh was dried. The entire process was carried out in a Petri dish with a piece of parafilm as a pure medium. Morphometric measurements were made using the iTEM/AnalySIS software.

The morphometry was performed manually due to the aggregation of the particles. The images were recalibrated based on the scale mark. A few images with the highest magnification and good quality were selected from each set. Since the particles had an oval or slightly elongated polygonal shape, their largest size was measured. Up to 20 distinguishable particles (if that many were available) were randomly selected from each processed image. The mean, standard deviation, and the smallest and largest particle sizes were calculated from the collected data (STATISTICA 13.0, StatSoft, Krakow, Poland).

### 2.4. Bacterial Cultures

The growth curves of 17 strains of lactic acid bacteria on nanoparticle-supplemented media (Table 1) were determined with the use of Bioscreen C (Labsystem, Helsinki, Finland) as in Gustaw et al. [27]. MRS media with the addition of the nanoparticles were prepared. In the study, four types of TiO_2_ from different manufacturers were used in four concentration variants (a, b, c, d), in triplicate. The experiment was performed for 72 h by measuring OD600 nm every 2 h. A control was performed in each experiment. The comparison of these values showed the inhibitory properties of the particular types of nanoparticles and their concentrations. On the basis of the results obtained, growth kinetics values and statistics were determined using the PYTHON script for individual strains and each medium variant used (Appendix A, Appendix A) [28].

## 3. Results

### 3.1. Characterization of E171/TiO_2_ NPs

#### 3.1.1. Zeta Potential of E171/TiO_2_ Nanoparticles

The zeta potential was noticed to be pH dependent (Figure 2). The curves had a similar course in all tested samples, analogous to those described in the literature [29,30]. The ζ values ranged from +40 mV (pH = 2) to −17 mV (pH = 10) for samples 1–3. In the case of sample 4, this range was slightly wider, i.e., from +44 mV to −22 mV, which suggested higher stability of dispersion. The determined isoelectric points (IEP) were pH = 7.8 for samples 1–3 and pH = 7.6 for TiO_2_ sample 4. The ζ values were positive below these pH values and negative above these values. 

#### 3.1.2. Transmission Electron Microscopy (TEM) Analysis of the Samples 

The particle size distribution was determined with the TEM technique, which also showed that all samples were in the range of nanomaterials. The distribution of the sizes of the three tested E171 samples (No. 1, 2, 3) were similar and ranged from 40 to 283 nm. Approximately 25–40% of the particles were smaller than 100 nm (Figure 3). The TiO_2_ NPs particle fraction (No. 4) was entirely in the nano-range (10–50 nm).

### 3.2. Bacteria

To check the impact of the nanoparticles, the growth of the 17 selected strains of lactic acid bacteria was monitored for 72 h using Bioscreen C.

The growth of a majority of the bacterial strains was inhibited on the medium supplemented with E171/TiO_2_ NPs, compared to the control (MRS medium). The differences in the growth inhibition between some strains were dependent on the type and concentration of TiO_2_ (Figure 4, Appendix A: Appendix A). The percentage of inhibition relative to the control was directly proportional to the increase in the concentration of the nanoparticles.

The smallest differences were found at the lowest concentration (60 mg/L) in most of the 17 strains tested. The growth curve on the medium supplemented with the lowest concentration of all TiO_2_ types (1, 2, 3, 4) was similar to the control in 12 strains, whereas the strains of *B. adolescentis* (Appendix A), *L delbrueckii* (Appendix A), *L. gasseri* (Appendix A), *L. fermentum* (Appendix A), and *L. intermedius* (Appendix A) exhibited growth inhibition even at such low concentrations of the nanoparticles. The differences in the growth curves between the E171/TiO_2_ NPs variants (1, 2, 3, 4) at this concentration were insignificant. Additionally, in the case of *L. delbrueckii* sp. *bulgaricus* (TiO_2_ NPs 4) (Appendix A), the onset of the exponential growth phase was significantly delayed, in comparison with the control.

At the two successive concentrations of 150 mg/L (b) and 300 mg/L (c), all strains were characterized by a certain degree of growth inhibition. Interestingly, both these concentrations sometimes induced a similar degree of bacterial growth inhibition, and the growth curves often overlapped. The higher concentrations also contributed to differences between the applied TiO_2_. This was evident in *L. rhamnosus*, *B. bifidum*, *L. acidophilus*, and *L. brevis* at the concentration of 150 mg/L (Appendix A) and in *B. longum*, *L. rhamnosus*, and *L. acidophilus* at the concentration of 300 mg/L (Appendix A).

Significant differences in bacterial growth were found in the majority of the tested strains cultured on the MRS medium supplemented with the four different types of TiO_2_, at the concentration of 600 mg/L (d) (Figure 4, Appendix A: Appendix A) (Table 2).

The highest growth inhibition was determined in *L. plantarum* and *B. adolescentis*, where virtually complete growth inhibition was demonstrated, and in *L. intermedius*, *L. fermentum*, *L. brevis*, *L. casei Lby*, *L. plantarum IB*, *B. bifidum*, and *L. rhamnosus* (Figure 4). It was also observed that the *B. adolescentis* and *L. helveticus* strains (at the concentration of 600 mg/L) were characterized by a delayed onset of the log phase, in comparison with the control (Figure 4 and Appendix A). Interesting findings were also obtained for the *L. gasseri*, *L. plantarum IB*., *L. rhamnosus*, and *L. helveticus* strains (Appendix A, and Figure 4), as the application of the different TiO_2_ types at the concentration of 600 mg/L resulted in differences in the growth of these bacteria, depending on the type of E171/TiO_2_ NPs.

The bacterial growth kinetics was calculated using the PYTHON script (Table 2). The highest inhibition of growth was caused by the concentration of 600 mg/L (d). In the case of the maximum OD, the lowest values (to 0.2 OD) were found at this concentration. Additionally, TiO_2_ No. 3 was found to cause the highest inhibition of bacterial growth in 11 strains, likewise TiO_2_ No. 2 in four strains (Table 2). In turn, in the case of max OD, the lowest inhibition was caused by nanoparticle No. 4. The longest cell doubling time was observed in six strains supplemented with TiO_2_ NPs (No. 4) and in five strains growing in the presence of TiO_2_ No. 2. The values of the Max Specific Growth Rate (1/h) indicate the length of the “exponential growth” phase. The shortest phase or its absence (no growth) was determined in the case of TiO_2_ No. 2 and 3. The analysis of the calculated values of the lag phase revealed prolonged duration of this growth stage even up to 51 h in 11 strains in the TiO_2_ NPs variant (No. 4), disregarding the lack of these data related to the absence of growth. The lag phase is usually extended, as the bacteria have to adapt to the new unfavorable environment, i.e., the nanoparticle-supplemented medium. However, in a few cases where the growth curves were relatively flattened, there were only slight differences in comparison with the control.

## 4. Discussion

The cultivation of the lactic acid bacteria in the presence of the nanoparticles showed the inhibition of bacterial growth; however, the concentration at which the minimal effect was noted was strain dependent. We showed that the lowest concentration that caused the growth inhibition in all strains was 150 or 300 mg/L. Interestingly, at these doses of the nanoparticles, there were evident differences in the bacterial response to the different E171/TiO_2_ NPs types used in the experiment. The number of works discussing this scientific topic is rather limited. Dudefoi et al. [4] reported that food-grade TiO_2_ particles did not significantly affect the human intestinal microbiota and showed a slight decline in the percentage of Gram-negative *B. ovatus* and an increase in the number of Gram-positive *C. cocleatum* strains. As reported by Ripolles-Avila et al. [31], depending on the dose, TiO_2_ NPs exhibit antibacterial activity against Gram-positive bacteria (*S. aureus*, *B. cereus*, *L. casei*, *L. bulgaricus*, *L. acidophilus*, and *L. lactis*). They showed that the optimal content of TiO_2_ nanoparticles in a bacterial culture suspension (100 μg/mL) reduced the amount of 2–3 log bacterial populations assessed after 24 h of incubation. Radziwił et al. [17] investigated interactions between TiO_2_ NPs (food-grade E171 and TiO_2_—P25) and bacteria ingested with food (e.g., *L. lactis*). They reported inhibition of bacterial growth (*L. lactis*, *L. rhamnosus*) induced especially by food-grade TiO_2_, as described in the present study as well. These authors suggested that E171 may have been trapped by food-borne bacteria in the intestine, which may have induced physiological changes in the most sensitive species. Lately, Mukherjee et al. [32] anatase (50 nm, 98% pure, hydrophilic) at 1 ppm significantly increased the growth of *Bacillus coagulans* after 14–18 h of incubation in the absence of light. No such effect in the presence of 0.1 ppm of TiO_2_ NPs was observed, up to 18 h of incubation, as compared with the control without NPs. The highest TiO_2_ concentration applied in the cited study (10 ppm) showed less pronounced effect than 1 ppm concentration, and the highest concentration was more or less similar to control. Authors also studied this NP at higher concentrations but reported efficient aggregation of nanoparticles that could result in the lack of interactions with bacteria. 

Authors point out that various factors can influence the interactions between NPs and intestinal bacteria, e.g., the surface charge of bacteria and nanoparticles, the surface charge of ingested food, the composition of the chemical substance, and the diet [33]. As reported by Pagnout et al. [23], the toxicity of TiO_2_ NPs is related to electrostatic interactions between bacteria and nanoparticles, leading to the adsorption thereof on the cell surface. Planchon et al. [34] supported the concept of the heterogeneity of bacterial populations. In their research, they evidenced that, after exposure to TiO_2_ NPs, some bacteria were fully coated by the compound, whereas a substantial part of the bacterial population was free from the nanoparticles, which resulted in differences in the metabolome and proteome. Similarly, Radziwił et al. [17] demonstrated that part of the bacterial population was free from TiO_2_ NPs, while some bacteria interacted strongly with NPs. 

Exposure of tissue to nanoparticles can have far-reaching consequences ex vivo as well as in vivo. In the ex vivo study involving a gastrointestinal tract model, Limage et al. [21] showed that the presence of commensal Gram-positive *L. rhamnosus* bacteria and nanoparticles changed the thickness and composition of the mucus layer. This is particularly disadvantageous, as it has been shown that *Lactobacillus* spp. can increase the production of mucins MUC2 and MUC3. With the mucus layer strengthened in this way, the attachment of enteropathogenic *Escherichia coli* is hampered, which provides protection against pathogen invasion [21,35].

An imbalance in the composition of the gut bacteria can cause some health disturbances as shown in several animal studies. Li et al. [36] reported changes in the intestinal microbiota composition and a significant decline in the *Bifidobacterium* count number in male mice receiving TiO_2_ NPs (1 mg/kg/day for 7 days). Pignet et al. [37] showed that orally administered TiO_2_ NPs (2 and 10 mg TiO_2_/kg b.w./day and 50 mg TiO_2_/kg b.w./day) had minimal effect on the composition of the intestinal microbiota in the mouse colon and small intestine but caused the reduced expression of the colonic mucin gene, increased expression of the β-defensin gene, colonic inflammation (decreased crypt length, infiltration of CD8+T cells, increased macrophages, increased expression of inflammatory cytokines).

Mu et al. [38] reported that administration of TiO_2_ NPs (10 and 50 nm) to young weaned mice for 2–3 months in the diet reduced the numbers of *Bifidobacterium* and *Lactobacillus*, which led to weight loss. In in vivo study conducted by Cao et al. [11], oral administration of TiO_2_ (E171, 112 nm) and TiO_2_ NPs (33 nm) to obese and non-obese mice (0.1% w/w in the diet for 8 weeks) resulted in a significant reduction in the intestinal amounts of *Bifidobacterium* and *Lactobacillus* bacteria count number. The authors found that TiO_2_ NPs induced more severe colon inflammation than TiO_2_ (E171), especially in the more susceptible obese mice, which was also associated with their high-fat diet. Authors suggested that TiO_2_ exposure of mice with reduced levels of *Bifidobacterium* and *Lactobacillus* may result in increased susceptibility to such diseases as irritable bowel syndrome [11]. 

## 5. Conclusions

Due to their unique physicochemical properties, titanium dioxide nanoparticles are produced all over the world. The high level of production and wide application of these nanoparticles create hazards to the environment and humans. Despite their widespread use as food additives, the risk of ingestion thereof has not been fully documented to date. Current research provides conflicting evidence of the effects of inorganic nanoparticles on the human microbiome. The application thereof as food additives and their subsequent impact on the function of the gastrointestinal tract, including a direct effect on the microbiota, require elucidation. The present study showed that bacterial growth was inhibited by both food-grade E171 and TiO_2_ NPs in most of the strains tested. This may suggest that the antimicrobial properties of NPs may alter the gut microbiota; therefore, further research is necessary in this field to understand the toxicity of NPs to the human microbiome.

## Figures and Tables

**Figure 1 foods-10-00939-f001:**
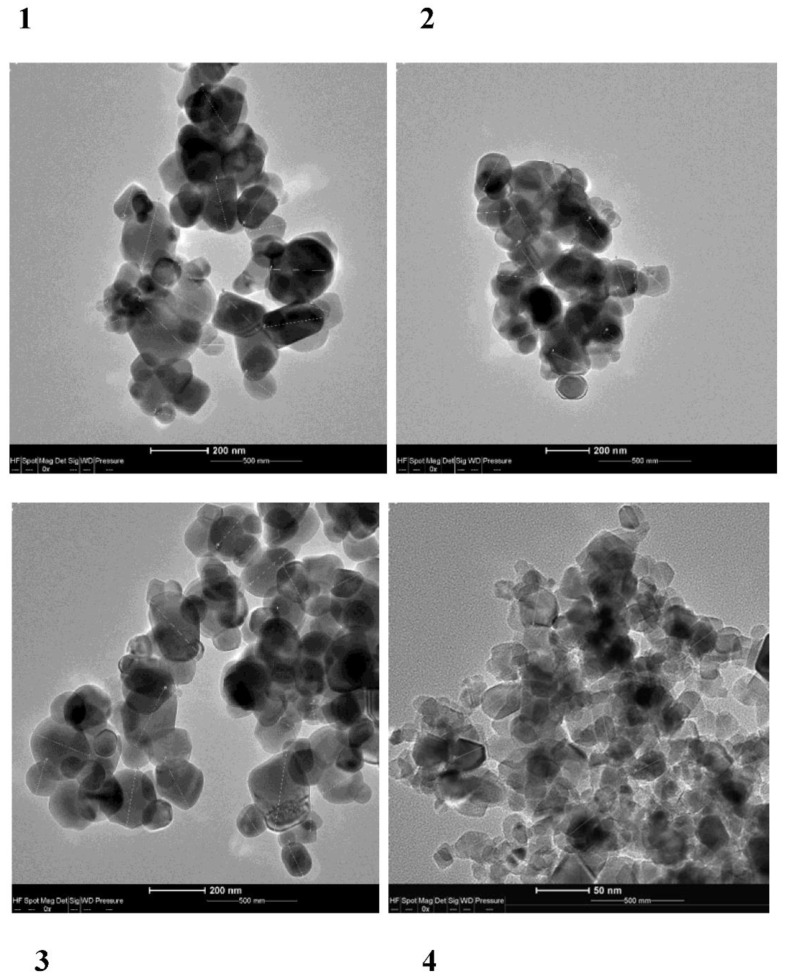
TEM images of TiO_2_. E171 (No. 1, 2, 3) and TiO_2_ NPs (No. 4).

**Figure 2 foods-10-00939-f002:**
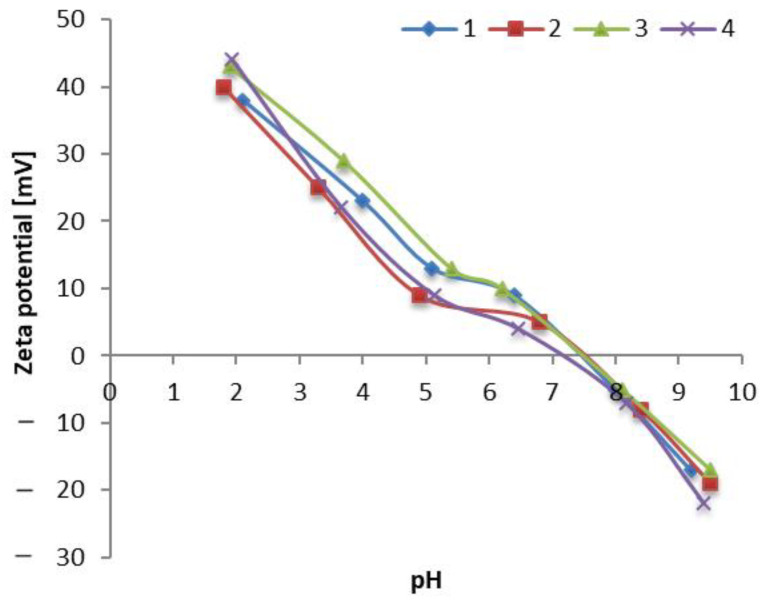
Zeta potential of the E171/TiO_2_ nanoparticles as a function of pH; E171 (No. 1, 2, 3), TiO_2_ NPs (No. 4).

**Figure 3 foods-10-00939-f003:**
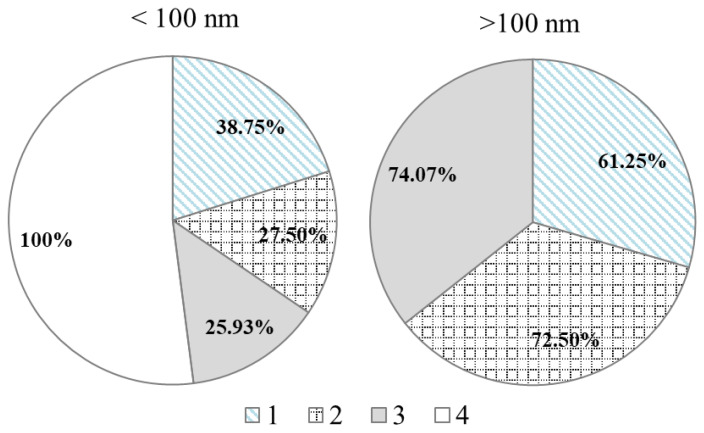
Size distribution of all tested TiO_2_ materials [%]; E171 (No. 1, 2, 3), TiO_2_ NPs (No. 4).

**Figure 4 foods-10-00939-f004:**
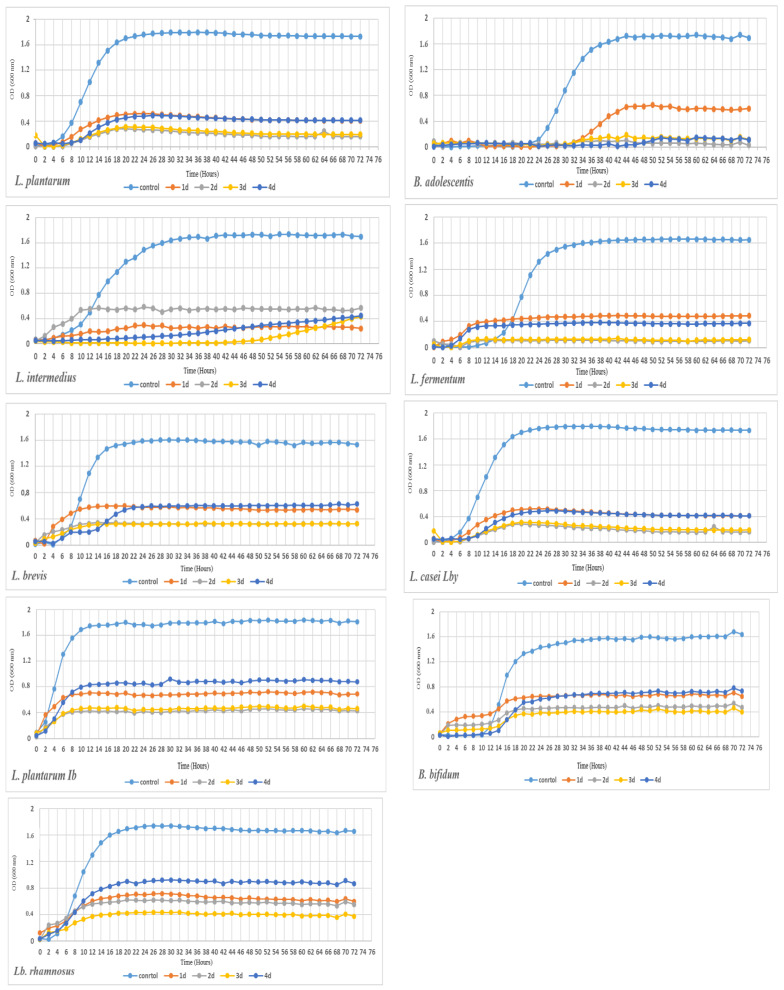
Growth of selected bacteria after application of four types of TiO_2_ at the concentration of 600 mg/L; E171 (No. 1, 2, 3), TiO_2_ NPs (No. 4).

**Table 1 foods-10-00939-t001:** List of bacterial strains under study.

Species and Strain
*Lacticaseibacillus rhamnosus* B-1445 *Lactiplantibacillus plantarum* IB *Bifidobacterium bifidum* B 41410 *Lactococcus lactis* PCM 2678 *Bifidobacterium adolescentis* DSM 20086 *Lactobacillus acidophilus* DSM 20079 *Bifidobacterium longum* B-41409. ATCC 15707. DSM 20,219 (intestine of adult hu man) *Pediococcus pentosaceus* *Lactobacillus johnsonii* DSM 10553 *Lacticaseibacillus casei Lby* *Lactobacillus delbrueckii* sp. *bulgaricus* *Lactobacillus gasseri* PCM 2500 *Limosilactobacillus fermentum* PCM 491 *Lactobacillus helveticus* DSM 20075 *Lactobacillus intermedicus* B 3693 *Levilactobacillus brevis* B 1139 *Lactobacillus plantarum* B 4496

**Table 2 foods-10-00939-t002:** Bacterial growth parameters (600 mg/L); E171 (No. 1, 2, 3), TiO_2_ NPs (No. 4).

Species	Types of TiO_2_	Lag Time (h)	Max Specific Growth Rate (h-1)	Doubling Time (h)	Max OD	Min OD	R2
*L. plantarum*	control	4.70	0.15	4.48	1.86	0.05	1.00
	1	1.36	0.04	16.65	0.64	0.02	0.99
	2	0.00	0.03	24.79	0.47	0.03	0.98
	3	0.00	0.03	24.47	0.54	0.05	0.96
	4	9.94	0.05	14.08	0.49	0.05	1.00
*B. adolescentis*	control	31.24	0.05	12.89	0.92	0.01	0.99
	1	34.21	0.07	10.11	0.65	0.00	1.00
	2	n.g	n.g	n.g	0.10	0.01	n.g
	3	n.g	n.g	n.g	0.19	0.02	n.g
	4	n.g	n.g	n.g	0.15	0.01	n.g
*L. intermedius*	control	8.54	0.08	9.00	1.63	0.07	1.00
	1	0.10	0.01	56.67	0.30	0.06	0.98
	2	0.79	0.06	11.92	0.59	0.05	0.98
	3	51.19	0.02	34.48	0.43	0.00	1.00
	4	20.55	0.01	83.28	0.44	0.05	1.00
*L. fermentum*	control	3.74	0.05	13.01	1.43	0.04	1.00
	1	16.21	0.16	4.45	1.67	0.01	1.00
	2	0.00	0.04	17.38	0.49	0,01	0.98
	3	n.g	n.g	n.g	0.12	0.01	n.g
	4	n.g	n.g	n.g	0.14	0.01	n.g
*L. brevis*	control	7.39	0.10	6.95	1.73	0.07	1.00
	1	1.38	0.07	9.33	0.60	0.02	0.99
	2	0.00	0.03	20.18	0.34	0.03	0.97
	3	0.00	0.03	23.54	0.34	0.02	0.99
	4	9.80	0.04	16.11	0.63	0.03	0.99
*L*. *casei Lby*	control	5.91	0.17	4.05	1.73	0.02	1.00
	1	6.51	0.05	13.50	0.52	0.01	1.00
	2	7.65	0.03	23.27	0.29	0.00	1.00
	3	9.83	0.03	20.44	0.31	0.01	0.91
	4	9.94	0.05	14.08	0.49	0.05	1.00
*L. plantarum IB*	control	14.55	0.24	2.84	1.83	0.01	1.00
	1	11.72	0.12	6.01	0.92	0.00	0.99
	2	10.47	0.13	5.15	1.32	0.10	1.00
	3	11.30	0.12	5.63	1.05	0.03	0.99
	4	9.93	0.11	6.09	1.18	0.00	1.00
*B. bifidum*	control	26.82	0.15	4.58	1.79	0.07	1.00
	1	0.00	0.03	23.45	0.71	0.05	0.97
	2	8.83	0.03	24.79	0.53	0.03	0.96
	3	11.68	0.03	22.35	0.46	0.06	0.98
	4	13.22	0.06	10.67	0.78	0.01	0.99
*L. rhamnosus*	control	9.24	0.16	4.21	1.75	0.05	1.00
	1	1.91	0.05	13.75	0.72	0.12	1.00
	2	0.00	0.05	13.07	0.62	0.02	0.98
	3	0.00	0.03	21.36	0.43	0.03	0.99
	4	2.85	0.08	9.10	0.92	0.03	1.00

n.g.—no growth.

## Data Availability

Not applicable.

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
