# Peer review of "Four Types of TiO2 Reduced the Growth of Selected Lactic Acid Bacteria Strains"

_foods, 2021, doi:10.3390/foods10050939_

Round 1
Reviewer 1 Report
This manuscript “Effect of TiO2 on lactic bacteria in the human gastrointestinal tract.” Investigates the response of strains of lactic acid bacteria to E171/TiO2 NPs, depending on their concentration, size, and applicability.
Recently some articles have been written about similar subjects:
Baranowska-Wójcik, E., Szwajgier, D., Oleszczuk, P. et al. Effects of Titanium Dioxide Nanoparticles Exposure on Human Health—a Review. Biol Trace Elem Res 193, 118–129 (2020). https://doi.org/10.1007/s12011-019-01706-6
Radziwill-Bienkowska JM, Talbot P, Kamphuis JBJ, Robert V, Cartier C, Fourquaux I, Lentzen E, Audinot JN, Jamme F, Réfrégiers M, Bardowski JK, Langella P, Kowalczyk M, Houdeau E, Thomas M, Mercier-Bonin M. Toxicity of Food-Grade TiO2 to Commensal Intestinal and Transient Food-Borne Bacteria: New Insights Using Nano-SIMS and Synchrotron UV Fluorescence Imaging. Front Microbiol. 2018 Apr 24;9:794. doi: 10.3389/fmicb.2018.00794. PMID: 29740421; PMCID: PMC5928251.
So, the topic is not new but, the article, contributes to positive and interesting new data of the topic in question and the introduction summarizes the previous works. Concerning the methodologies used, I think they were appropriated, and the results obtained are interesting. Conclusions are supported by the results.
Nevertheless, I have some comments:
Page 5, lines 166-179. I am confused…these sentences do not belong to the article!!! Do they??
Page 5, lines 190-192. The same…the sentence does not belong to the work…
So, I propose a minor revision of the text of the work before acceptance.
Author Response
Dear Reviewer
We appreciate the time and effort that you and the reviewers devoted in order to provide the feedback and we are grateful for the insightful comments and valuable improvements concerning our paper. We have corrected the paper according to all suggestions of the reviewers. The changes are highlighted in the manuscript.
I remain respectfully yours.
Here is a point-by-point response to the review:
1# Comments and Suggestions for Authors
Page 5, lines 166-179. I am confused…these sentences do not belong to the article!!! Do they??
AU: Thank you for pointing this out. It was corrected as suggested. Lines 166-179 were removed from the text.
Page 5, lines 190-192. The same…the sentence does not belong to the work…
AU: Thank you for pointing this out. I'm sorry, this is my oversight. It was corrected as suggested. Lines 190-192 were removed from the text.
Let me thank you for your valuable comments concerning my paper.
Reviewer 2 Report
The manuscript entitled “Effect of TiO2 on lactic bacteria in the human gastrointestinal tract” needs to be revised before it can be accepted for publication in “Foods”. Some recommendations are as the following:
- As the authors presented in the discussion part, several previous studies investigated the interaction between TiO2 nanoparticles and intestinal bacteria. Therefore, the novelty of the current study is not clear. Please clearly indicate the difference between this manuscript and the ones that are already present in the literature.
- In the materials and methods section, the statistical analysis is only described for the microscopy analysis. Please clearly indicate the number of measurements and other related statistical information for all assays.
- Lines 166-180 and 190-192: Please remove the template sentences. This indicates that the manuscript is prepared without paying attention and not checked before submission.
Author Response
Dear Reviewer
We appreciate the time and effort that you and the reviewers devoted in order to provide the feedback and we are grateful for the insightful comments and valuable improvements concerning our paper. We have corrected the paper according to all suggestions of the reviewers. The changes are highlighted in the manuscript.
I remain respectfully yours.
Here is a point-by-point response to the review:
- As the authors presented in the discussion part, several previous studies investigated the interaction between TiO2 nanoparticles and intestinal bacteria. Therefore, the novelty of the current study is not clear. Please clearly indicate the difference between this manuscript and the ones that are already present in the literature.
AU: Due to the huge diversity of microbiota in every human being, it is not easy to fully or even largely characterize the gut microbes. In this article we focused on the checking of the influence of titanium dioxide and titanium nanoparticles on a significant number of strains. Due to laboratory limitations, 17 strains from 3 genera were selected to start this topic. The work using next strains is in progress. Therefore, the novelty of our work is the characterization of strains that have not been previously characterized in this scientific context, Please also note that there are only 3-4 other studies similar to our research (all discussed in our paper). We pointed out in Discussion chapter the differences in results obtained in our and cited papers. The cited papers studied other bacterial strains and our work fits very well in this context and reduces the lack of results in this topic. However, as our Dept. of Biotechnology, Microbiology and Human Nutrition has dozens of intestinal lactic acid bacteria strains in the collection, we plan to gradually replenish the knowledge on this topic.
- In the materials and methods section, the statistical analysis is only described for the microscopy analysis. Please clearly indicate the number of measurements and other related statistical information for all assays.
AU: Thank you for your comment. Information about the number of measurements performed has been added in the " Bacterial cultures" section, as well as we provided citation of the statistical method. In addition, statistical values calculated from the Bioscreen experiment have been included in Table 2 as well as in the supplementary materials. The cited paper (Hoeflinger et al [28]), describes the calculations and provides a PYTHON script.
- Lines 166-180 and 190-192: Please remove the template sentences. This indicates that the manuscript is prepared without paying attention and not checked before submission.
AU: Thank you for pointing this out. I'm sorry for this oversight. It was corrected as suggested. Lines 166-180 and 190-192 were removed from the text.
Let me thank you for your valuable comments concerning my paper.
Reviewer 3 Report
The authors in the manuscript “Effect of TiO2 on lactic bacteria in the human gastrointestinal tract” should be revised to focus on the experiments of the current study and demonstrated the effect in the human gastrointestinal tract. The title is too general. It does not reflect the works carried out in the study. It is ambiguous to use the phrases like human gastrointestinal tract. The objective is not clear and concise. They didn’t include details in the methods, for example the bacterial conditions of Bifidus, that they are facultative anaerobic. The discussion section was focusing on other reports and not in the results found in this study.
To detail:
The authors used microflora that is an ancient term.
It is correct to refer to the full name of species the first time they appear throughout the text and use the generally accepted abbreviated form (Staphylococcus aureus 1st, then S. aureus, for example).
It is not correct to write Lb, rhamnosus (appears often). Also, they need to use the new nomenclature described by Zheng et al., Int. J. Syst. Evol. Microbiol. DOI 10.1099/ijsem.0.004107
L103. Figure 1, the need to describe the letter A, B, C and D.
L07. Change min instead minutes
L166-180, L190-192. What is that? The template?
L196. To determine the particle size, the authors could use the Zetasizer 3000 instrument?
L208. Change h instead hours as
L222. Change L. delbrueckii instead L. duberulecki
Author Response
Dear Reviewer
We appreciate the time and effort that you and the reviewers devoted in order to provide the feedback and we are grateful for the insightful comments and valuable improvements concerning our paper. We have corrected the paper according to all suggestions of the reviewers. The changes are highlighted in the manuscript.
I remain respectfully yours.
Here is a point-by-point response to the review:
The authors in the manuscript “Effect of TiO2 on lactic bacteria in the human gastrointestinal tract” should be revised to focus on the experiments of the current study and demonstrated the effect in the human gastrointestinal tract. The title is too general. It does not reflect the works carried out in the study. It is ambiguous to use the phrases like human gastrointestinal tract. The objective is not clear and concise. They didn’t include details in the methods, for example the bacterial conditions of Bifidus, that they are facultative anaerobic. The discussion section was focusing on other reports and not in the results found in this study.
AU:
-The title was changed in order to reflect the content of this work: “Four types of TiO2 reduced the growth of selected lactic acid bacteria strains.”
-The aim of this work was worded more precisely in the abstract as well as in the last sentence of the Introduction, line: 73
-“They didn’t include details in the methods, for example the bacterial conditions of Bifidus, that they are facultative anaerobic”.
AU: Yes, it is true, regarding anareobic conditions we adapted the experimental conditions from one of our previous articles that contain a detailed description of these methods: “Gustaw. K., Michalak. M., Polak-Berecka. M., & WaÅ›ko. A. (2018). Isolation and characterization of a new fructophilic Lactobacillus plantarum FPL strain from honeydew. Annals of microbiology, 68(7), 459-470. doi:10.1007/s13213-018-1350-2.”
- The discussion section was focusing on other reports and not in the results found in this study.
AU: Discussion was completely rebuilt according to the suggestions of the reviewer. The improved discussion, in the first part, contains results works published in the past, containing similar results to ours. Then come results containing ex vivo, and, at the end, in vivo results concerning the effect of TiO2 on the growth of lactic acid bacteria. We are aware that the works discussed in the first paragraph of the discussion are directly related to our results, but as you can see, there is really a limited number of such works (we found only 4 papers). Therefore, we thought it will be useful to present the works concerning also in vivo studies. If you find them unnecessary, we will remove the in vivo works, however, they may be interesting for the reader because they show the effect on the higher organism.
AU: Editing errors were corrected in the text according to the Reviewer’s remarks (marked with green color)
The authors used microflora that is an ancient term.
AU: It was corrected as suggested. The microflora is replaced by microbiota. Line: 20, 58, 59, 71, 264, 300, 303, 327
It is correct to refer to the full name of species the first time they appear throughout the text and use the generally accepted abbreviated form (Staphylococcus aureus 1st, then S. aureus, for example).
It is not correct to write Lb, rhamnosus (appears often). Also, they need to use the new nomenclature described by Zheng et al., Int. J. Syst. Evol. Microbiol. DOI 10.1099/ijsem.0.004107
AU: It was corrected as suggested. We used the new nomenclature in manuscript and supplementary material. Line: 147, 148, 156, 159, 162, 211, 229, 233
L103. Figure 1, the need to describe the letter A, B, C and D.
AU: It was corrected as suggested. On Figure 1 letters A, B, C and D have been changed to 1, 2, 3, 4 according to the legend.
L07. Change min instead minutes
AU: It was corrected as suggested. Line: 106, 121
L166-180, L190-192. What is that? The template?
AU: Thank you for pointing this out. I'm sorry for my oversight. It was corrected as suggested. Lines 166-180 and 190-192 were removed from the text.
L196. To determine the particle size, the authors could use the Zetasizer 3000 instrument?
AU: No, the size of the nanoparticles was estimated using a TEM transmission electron microscope (FEI Tecnai G2 T20 X-Twin Ltd., Japan). Line: 117
L208. Change h instead hours as
AU: It was corrected as suggested. Line: 190, 243, 270, Table 2
L222. Change L. delbrueckii instead L. duberulecki
AU: It was corrected as suggested. Line: 200
Let me thank you for your valuable comments concerning my paper. Amendments in the paper according to your suggestions are marked with green.
Round 2
Reviewer 2 Report
The authors addressed all the points that are raised by the reviewer. The manuscript is now improved.
Reviewer 3 Report
No comments